**Data Availability Statement:** All relevant data are within the manuscript and its Supporting Information files

# The accuracy of healthcare worker versus self collected (2-in-1) Oropharyngeal and Bilateral Mid-Turbinate (OPMT) swabs and saliva samples for SARS-CoV-2

**Seow Yen Tan**[1]☯*, **Hong Liang Tey**[2]☯, **Ernest Tian Hong Lim**[3]☯, **Song Tar Toh**[4]☯, **Yiong Huak Chan**[5], **Pei Ting Tan**[6], **Sing Ai Lee**[7], **Cheryl Xiaotong Tan**[8], **Gerald Choon Huat Koh**[9‡], **Thean Yen Tan**[10‡], **Chuin Siau**[11‡]

1 Department of Infectious Diseases, Changi General Hospital, Singapore, Singapore, 2 Department of Dermatology, National Skin Centre, Singapore, Singapore, 3 Emergency Department, Woodlands Health Campus, Singapore, Singapore, 4 Department of Otorhinolaryngology- Head and Neck Surgery, Singapore General Hospital, Singapore, Singapore, 5 Biostatistics Unit, Yong Loo Lin School of Medicine, Singapore, Singapore, 6 Clinical Trials and Research Unit, Changi General Hospital, Singapore, Singapore, 7 Sheares Healthcare Group Pte Ltd, Singapore, Singapore, 8 Temasek International Pte Ltd, Singapore, Singapore, 9 MOH Office for Healthcare Transformation, Singapore, Singapore, 10 Department of Laboratory Medicine, Changi General Hospital, Singapore, Singapore, 11 Department of Respiratory & Critical Care Medicine, Changi General Hospital, Singapore, Singapore

☯ These authors contributed equally to this work.
‡ These authors are joint senior authors on this work.
* tan.seow.yen@singhealth.com.sg

## Abstract

### Background

Self-sampling for SARS-CoV-2 would significantly raise testing capacity and reduce healthcare worker (HCW) exposure to infectious droplets personal, and protective equipment (PPE) use.

### Methods

We conducted a diagnostic accuracy study where subjects with a confirmed diagnosis of COVID-19 (n = 401) and healthy volunteers (n = 100) were asked to self-swab from their oropharynx and mid-turbinate (OPMT), and self-collect saliva. The results of these samples were compared to an OPMT performed by a HCW in the same patient at the same session.

### Results

In subjects confirmed to have COVID-19, the sensitivities of the HCW-swab, self-swab, saliva, and combined self-swab plus saliva samples were 82.8%, 75.1%, 74.3% and 86.5% respectively. All samples obtained from healthy volunteers were tested negative. Compared to HCW-swab, the sensitivities of a self-swab sample and saliva sample were inferior by 8.7% (95%CI: 2.4% to 15.0%, p = 0.006) and 9.5% (95%CI: 3.1% to 15.8%, p = 0.003) respectively. The combined detection rate of self-swab and saliva had a sensitivity of 2.7% (95%CI: -2.6% to 8.0%, p = 0.321). The sensitivity of both the self-collection methods are higher when the Ct value of the HCW swab is less than 30. The specificity of both the self-swab and saliva testing was 100% (95% CI 96.4% to 100%).

**Funding:** This study was funded by Sheares Healthcare Group Pte Ltd. The funder provided support in the form of salaries for authors SAL and CXT, but did not have any additional role in the study design, data collection and analysis, decision to publish, or preparation of the manuscript. The specific roles of these authors are articulated in the 'author contributions' section. Besides that, author CXT is employed by Temasek International Pte Ltd, and was acting on behalf of Sheares Healthcare Group Pte Ltd for the study. Temasek International Pte Ltd did not have any additional role in the study design, data collection and analysis, decision to publish, or preparation of the manuscript.

**Competing interests:** Author CXT is employed by Temasek International Pte Ltd, and was acting on behalf of Sheares Healthcare Group Pte Ltd for the study. This commercial affiliation does not alter our adherence to PLOS ONE policies on sharing data and materials.

## Conclusion

Our study provides evidence that sensitivities of self-collected OPMT swab and saliva samples were inferior to a HCW swab, but they could still be useful testing tools in the appropriate clinical settings.

## Introduction

The current "gold standard" for testing for SARS-CoV-2 requires health care workers to collect a nasopharyngeal (NP) sample from a patient. NP sampling is very uncomfortable for the patient and requires deployment of trained personnel and use of personal protective equipment (PPE) which are in limited supply.

A prior study has shown that a combination of oropharyngeal and anterior nares swabs is equivalent in sensitivity to an NP swab in 190 ambulatory symptomatic patients [1]. In another study on 236 ambulatory subjects, the performance of self-collected nasal and throat swabs is at least equivalent to that of health worker collected swabs for the detection of SARS-CoV-2 and other respiratory viruses [2].

The international community is actively searching for an even less invasive means of sample collection: saliva. In a recent study by Yale University on 29 subjects [3], it was suggested that a large volume sample of saliva collected from COVID-19 inpatients can be more sensitive than NP swabs for SARS-CoV-2 detection, and saliva samples had significantly higher COVID-19 viral titres than NP swabs (p = 0.001). Furthermore, the same study showed that sensitivity of COVID-19 in saliva was more consistent throughout extended hospitalization compared to NP swabs.

In addition, there are a number of studies done on saliva testing for COVID-19 which have shown promising results, reporting 91.7%, and 100% positivity in saliva samples of COVID-19 patients [4, 5]. Iwasaki et al found an overall concordance rate of 97.4% for COVID-19 detection with a strong concordance between NP swabs and saliva sampling ($\kappa = 0.874$) among 66 COVID-19 negative and 10 COVID-19 positive subjects [6]. Furthermore, a study done by To et al. showed that viral RNA could still be detected in saliva samples in a third of their twenty-three patients 20 days or longer after symptoms onset despite the development of COVID-19 antibodies [7]. A meta-analysis conducted on 26 saliva studies also showed a positive detection rate of 91%, comparable to the detection rate of 98% from nasopharyngeal swabs [8]. All these studies had small sample sizes (all <30 COVID-positive subjects) and only one study also sampled COVID-negative subjects.

It is still currently unknown whether a self-collected combined Oropharyngeal and Bilateral Mid-Turbinate (OPMT) sample, or a self-collected saliva sample is equivalent to a swab done by a health care worker (HCW). If the self-collection of samples is proven to be a reliable alternative to a HCW swab, it would reduce the reliance of trained personnel to collect samples and enable a rapid increase in testing capacity. It would also reduce greatly the biosafety risk that is posed to HCWs and help with PPE conservation efforts.

## Materials and methods

### Study design and trial oversight

This was a prospective study involving 401 subjects who were previously tested positive for COVID-19 by RT-PCR, and 100 healthy volunteers. This study was approved by the

SingHealth Centralised Institutional Review Board. Written informed consent was obtained from the subjects.

## Participants

The first group consisted of patients who were confirmed to have COVID-19, and who were cared for in either a hospital (Changi General Hospital), or a community care facility (Community Care Facility @ EXPO). Diagnosis of COVID-19 was confirmed via a positive RT-PCR from a nasopharyngeal swab. The subjects in this group were recruited within 3 days of admission to the study site and they were recruited from 31 May 2020 to 10 June 2020. The patients who were eligible were approached directly at the study site, and were invited to participate in the study, and the study procedures were carried out on the same day. Recruitment was carried out until the target sample size was achieved. At the time of the study, the majority of COVID-19 cases belong to the migrant worker population, which primarily consisted of healthy young male adults, mainly from Bangladesh and India. Hence, this group of subjects is not representative of the general population in Singapore.

Inclusion criteria applicable to this group include:

- Male and female patients, $\geq$ 21 years-old

- Tested positive for COVID-19

- Admitted to study site within the previous 3 days

- Ability to provide informed consent

- Compliance with all aspects of study protocol, methods and provision of samples

- Ability to read and understand English

Exclusion criteria applicable to this group include:

- Nosebleeds in past 24 hours

- Previous nasal surgery in past 4 weeks

- Acute facial trauma within 8 weeks

- Unable to demonstrate understanding of study and instructions

- Experienced severe adverse reactions on prior nose and/or throat swabs

- Not willing to have all 3 samples collected

The second group comprised 100 healthy volunteers who were asymptomatic and well on the day of the study, with no recent COVID-19 exposure. This was done on 18 and 19 July 2020. The study subjects were recruited via an open advertisement.

Inclusion Criteria for this group include:

- Males and females, $\geq$ 21 years-old

- Ability to provide informed consent

- Capable of understanding and complying with the requirements of the study

- Ability to read and understand English

Exclusion Criteria applicable to this group were:

- Displaying symptoms of an acute respiratory infection

- Known close contact with an individual diagnosed with COVID-19 within the last 3 months

- Previously diagnosed with COVID-19

- Nosebleeds in past 24 hours

- Previous nasal surgery in past 4 weeks

- Acute facial trauma within 8 weeks

- Unable to demonstrate understanding of study and instructions

- Experienced severe adverse reactions on prior nose and/or throat swabs

- Not willing to have all 3 samples collected

## Test procedures

Study subjects underwent three sequential test sample collection procedures within one study visit in the following order:

1. Each subject self-collected a sample combining OP and bilateral MT swabs using a single swab stick;

2. A trained healthcare worker then collected a combined OP and bilateral MT swab using another single swab stick;

3. The subject then self-collected a saliva sample.

Study subjects were shown instructional videos for both the OPMT self-swab and saliva collection prior to commencing the test procedures. Study team members were present on site to observe and supervise the self-collection process. Posterior oropharyngeal saliva, commonly described as deep throat saliva was collected for this study. Synthetic fibre swabs were used for collection of the OP and MT samples by both subject and healthcare worker, and immediately placed in universal transport medium (UTM), while saliva samples were collected using the SAFER-Sample™ (by Lucence Diagnostics). All samples were double bagged and stored at air-conditioned room temperature in a chiller bag and transported to assigned laboratory on the same day. Upon arrival in the laboratory, they were stored at 2˚C to 8˚C. All samples were processed with 24 hours of sample collection.

Nucleic acid extraction was performed using PerkinElmer Nucleic Acid Extraction Kits (KN0212) on the Pre-Nat II Automated Workstation (PerkinElmer®, United States), Extraction of swab samples followed the indicated protocol for oropharyngeal swabs, while extraction of saliva samples followed a protocol consisting of pre-liquefaction with dithiothreitol (protocol attached in S1 File). Reverse transcription polymerase chain reaction (RT-PCR) was performed on the Quantstudio™ 5 Real Time PCR system (Thermo Fisher, United Kingdom) using the PerkinElmer® SARS-CoV-2 Real-time RT-PCR Assay. The targets were the 'N' gene and 'ORF1ab' gene. There is an internal control target that is present in every RT-PCR reaction. The cycle threshold (Ct) values of the 'N' gene were used in the analysis involving Ct values.

## Outcomes

The primary objective of the study was to evaluate the accuracy of self-collected (2-in-1) OPMT swabs and self-collected saliva samples for SARS-CoV-2 versus that of HCW-collected (2-in-1) MT and OP swabs. The secondary objective was to evaluate the correlation of PCR Ct values of self-collected saliva samples and swabs with comparator healthcare worker-collected swabs.

## Sample size

Firstly, we postulated that OPMT self-swabbing was as accurate as HCW-obtained swabs. Postulating a 100% accuracy, 400 subjects will be required to achieve a lower 95% confidence interval 99.1% (which gives a less than 1% error rate). With the computed sample size of 400 subjects, a non-inferiority could be achieved with at most a 7% difference for OPMT self-swabbing compared to the HCW-obtained swabs. If the study included only subjects who were diagnosed with COVID-19, all positive results would be regarded as true positives. Hence, to address that gap in the form of specificity of self-swabs and saliva testing in the diagnosis of COVID-19, a further study on 100 healthy subjects was conducted. The hypothesis was that with 100% accuracy, the error rate for a false negative was 3.6%.

## Statistical analysis

All analyses were performed using SPSS 25.0 with statistical significance set at $p < 0.05$.

The estimates for the positivity results of the 3 methods were presented as numbers and percentages. The differences with 95% confidence interval (CI) between self-collection methods and HCW-obtained swabs to assess for non-inferiority was calculated. Sensitivity and specificity of the two self-collection methods were compared with HCW-obtained swabs and results were stratified by Ct values. Spearman's test was used to assess the correlation of the PCR Ct values across the 3 groups.

# Results

A total of 401 COVID-19 positive and 100 COVID-19 negative subjects were recruited. Of the 401 COVID-19 positive subjects, 23 were recruited from Changi General Hospital, and 378 were recruited from the community care facility @ Expo. The symptomatic COVID-19 positive subjects that were recruited were well patients, whose clinical presentation was that of an upper respiratory tract infection. None of the subjects required oxygen supplementation.

Only the demographic data of subjects from Changi General Hospital was known. The full demographic data of the subjects that were admitted to the community care facility could not be made available to us due to prevailing regulations of the study site during the period when the study was conducted, hence we do not have the data of the age of the subjects that were admitted to the community care facility. However, we were able to surmise that the age range of patients admitted to the community care facility was 21 to 45, due to the admission criteria to the facility, and the inclusion criteria for the study. A summary of the profile of recruited subjects are listed in Tables 1 and 2 below.

All subjects went through the test procedures—500 participants (400 COVID-19 positive, 100 COVID-19 negative) were able to provide all 3 samples, and one subject was unable to provide a saliva sample despite a prolonged attempt. All participants tolerated the test procedures well and did not experience any adverse events.

In the group of subjects who were COVID-19 positive, twenty-seven (6.7%) patients were tested negative across all 3 samples. This may be explained by the fact that they are recovering and viral shedding may have ceased at point of testing. Forty-two (10.5%) subjects reported ≥1 symptom of acute respiratory infection (ARI) (e.g. fever, cough, rhinorrhoea, sore throat, malaise) on the day of study recruitment while 371 (92.5%) subjects reported being within 7 days from onset of COVID-19 illness.

The detection rates of the HCW swab, self-swab, saliva, and combined self-swab plus saliva samples were 82.8%, 75.1%, 74.3% and 86.5% respectively (**Table 3**). Compared to HCW-swabs, the detection rate was lower for self-swab by 8.7% (95% confidence interval, CI = 2.4% to 15.0%, p = 0.006) and for saliva samples by 9.5% (95%CI = 3.1% to 15.8%, p = 0.003). When

**Table 1. Profile of COVID-19 positive subjects (N = 401).**

|  | N (%) |
|---|---|
| Age, years |  |
| Min–max | 21–54 |
| Mean (SD)* | 37.26 (6.4) |
| Male | 401 (100.0) |
| Presence of symptoms on study day |  |
| No | 359 (89.5) |
| Yes | 42 (10.5) |
| Duration between illness onset to study day, days |  |
| Range | 1–25 |
| Mean (SD) | 5.65 (2.1) |
| Days from first positive swab to study day, days |  |
| Range | 1–20 |
| Mean (SD) | 5.48 (1.8) |

* Calculation based on the known age of 24 subjects.

the results of both the self-swab and saliva testing were combined, the detection rate was higher by 2.7% (95%CI = -2.6% to 8.0%, p = 0.321) but this was not statistically significant.

The sensitivities of the self-swab, saliva and combined self-swab plus saliva testing, when compared to the HCW swab were 83.6%, 80.6% and 92.3% respectively. Table 4 shows the contingency tables comparing the HCW swab vs self-swab; HCW swab vs saliva, and HCW vs combined self-swab plus saliva respectively.

Using the Ct values ('N' gene) of HCW swabs as reference, 3 categories of Ct values (i.e. <25, 25–30 and >30) were studied. It was observed that the sensitivity of self-swab (**Table 5**) and saliva testing (**Table 6**) performed better at the lower Ct values, suggesting that the sensitivity of self-collection methods approaches to that of HCW swab, when the viral load was higher.

There was a good correlation of PCR Ct values between self-swab and HCW swab (r = 0.825, p<0.001) but moderate correlation between saliva samples and HCW swab (r = 0.528, p<0.001). The self-swab has a better agreement with the HCW swab. Using Wilcoxon Signed Rank Test, the difference in CT values between self-swab and HCW swab is statistically significant, where p = 0.026. Similarly for the saliva and HCW swab, where p<0.001. Figs 1 and 2 show the scatterplot of the correlation between the Ct values of the HCW swab and the self-swab as well as the saliva respectively. Table 7 shows the distribution of the Ct values of the 3 tests.

One hundred healthy volunteers were recruited, and all of them were able to provide the 3 required samples. All the samples obtained from the healthy volunteers were tested negative for SARS-CoV2. This implies that the specificity of the self-swab and saliva sampling was 100% (95% CI 96.4% to 100%) with an error rate of 3.6% for having a false negative.

**Table 2. Profile of COVID-19 negative subjects (N = 100).**

| Gender | N (%) |
|---|---|
| Female | 51 (51.0) |
| Male | 49 (49.0) |
| Age (years) |  |
| Mean (SD) | 38.24 (10.16) |
| Range | 22–70 |

**Table 3. Detection rates of various modalities in all subjects.**

|  | HCW Swab | Self-Swab | Saliva | Self-Swab + Saliva |
|---|---|---|---|---|
| Count | 336 | 301 | 297 | 347 |
| Percentage | 83.8% | 75.1% | 74.3% | 86.5% |
| 95% CI | 79.8% - 87.3% | 70.1% - 79.2% | 69.7% - 78.5% | 82.8% - 89.7% |

## Discussion

This study shows that the sensitivity of a self-swab or saliva sample on its own is lower than HCW swab. However, the sensitivity of a combined self-swab and saliva collection is equivalent to that of a HCW swab. Another significant finding is that the self-swab and saliva samples have a higher sensitivity when the viral load is higher, and this generally occurs during the early stages of COVID-19. The sensitivity of both self-swab and saliva testing drops significantly when the Ct values of the HCW swab is more than 30. A study from Singapore [9] reported that viral cultures were negative from samples with Ct values $> = 30$ (i.e. when viral load is low), and the SARS-CoV-2 virus often cannot be isolated or cultured after day 11 of illness [10]. Thus, the results of this study support the use of self-testing methods as a replacement for a HCW swab in the early phase of COVID-19 illness when viral loads are high, and the sensitivities of the self-swab and saliva are similar to that of the HCW swab.

The strength of our study is the large number of subjects confirmed to have COVID-19. Besides that, the study also included a high proportion of asymptomatic individuals who were picked up because of Singapore's proactive mass screening policy. The combination of self-swab and saliva sampling performed well in these asymptomatic subjects, implying that the strategy of combined self-testing, has the ability diagnose COVID-19 in asymptomatic individuals with a sensitivity equivalent to that of a swab by a HCW. The study results from the healthy volunteers indicate a low false positive rate with self-collection methods.

These findings, indicate that self-collection methods may be a useful tool for COVID-19 surveillance in the asymptomatic individuals, and in situations where testing capacity needs to be scaled up rapidly, without a need for large increase of manpower, and without increased infectious exposure to the swabbing staff. Testing strategies can be tailored based on the target population and the intended use of the various tests on its own or in combination.

**Table 4. Comparison between HCW swabs and the self-swab/saliva.**

|  | HCW Swab | | |
|---|---|---|---|
|  | Not detected | Detected | p value[*] |
| Self-swab |  |  |  |
| Not detected | 45 | 55 | <0.001 |
| Detected | 20 | 281 (83.6) |  |
| Saliva |  |  |  |
| Not detected | 37 | 65 | <0.001 |
| Detected | 27 | 270 |  |
| Self-swab plus saliva |  |  |  |
| Not detected | 27 | 26 | 0.207 |
| Detected | 37 | 310 |  |

[*] p value was obtained from McNemar test

**Table 5. Sensitivity of self-swab, stratified by Ct values of HCW swab.**

| HCW Swab Ct | Number of subjects | Sensitivity |
|---|---|---|
| <25 | 60 | 100% (94.0–100) |
| 25–30 | 81 | 96.3% (89.6–99.2) |
| >30 | 195 | 73.3% (66.5–79.4) |

The way the study findings were presented are unlike most studies involving saliva testing for COVID-19. This is probably due to the fact that our study is carried out on subjects who are already known to have COVID-19, unlike most studies which are done in testing centres where the potential subjects' results are still unknown. This also meant that the sampling was done later in the subjects' trajectory of illness, as they were first tested positive for COVID-19, then enrolled into the study. The later sampling possibly had a negative impact on the sensitivity of the saliva [11].

Another key study limitation, is that the demographics of the COVID-19-positive population was skewed, consisting solely of male migrant workers, the worst affected group of the pandemic in Singapore, at the time this study was conducted. Hence the results from this study might not be applicable to the general population, without the inclusion of paediatric and elderly population segments. The migrant worker population in this study, which consist of generally young and healthy males, is also not representative of the demographics of Singapore.

The addition of the stabilising solution to the deep throat saliva sample, could have also decreased the yield of the saliva testing. Studies utilising saliva test kits that do not require the addition of stabilising fluid generally report equivalent sensitivities of the saliva test when compared to a HCW swab [3, 12]. Hence the use of stabilising solution is a key consideration in future design of saliva test kits.

The study team members observed that, despite clear instructions, many subjects still needed guidance with the self-collection methods. For the self-swab, the most commonly encountered scenario was that, the subjects needed guidance in breaking the swab stick. The saliva collection presented a greater challenge to the subjects. The flow of saliva from the funnel into the collection container was not smooth, and the additional step of adding the stabilising fluid required prompting. These necessitated the presence of a trained staff to troubleshoot and ensure that the correct steps are carried out. We believe that these observations are useful in the re-design of collection containers to enhance results and end users' acceptability. Both the self-swab and saliva collection require dexterity and this would limit its applicability in segments of the population who are not able to do so.

We caution against widespread, unsupervised implementation of self-collection methods. The reliability and effectiveness of self-collection methods may also be dependent on social and economic drivers, hence potentially influencing the test performance. For example, individuals who face a potential loss of income or unemployment if tested positive or travellers having a test done at immigration clearance may deliberately do a suboptimal self-test to influence the test outcome.

**Table 6. Sensitivity of saliva, stratified by Ct values of HCW swab.**

| HCW Swab Ct | Number of subjects | Sensitivity |
|---|---|---|
| <25 | 60 | 96.7% (88.5–99.6) |
| 25–30 | 81 | 92.6% (84.6–97.2) |
| >30 | 194 | 70.6% (63.7–76.9) |

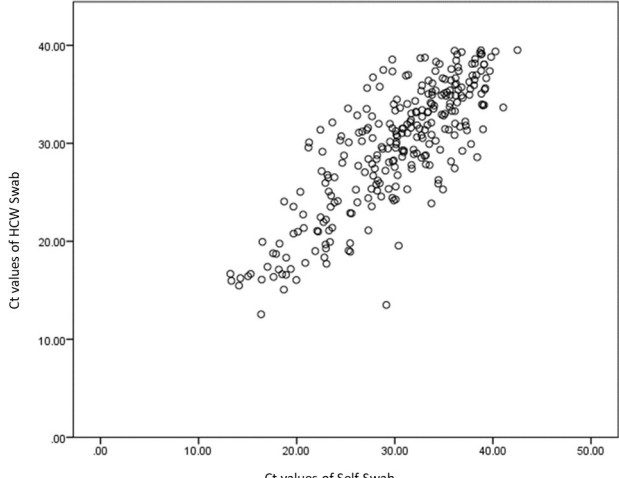

**Fig 1. Correlation of Ct values of HCW swab and self-swab.**

Hence, it is important to have designated personnel to supervise the self-collection process, ensuring that the correct test procedures are carried out. These personnel need not be a HCW and the supervision process will have a lower exposure risk (supervisor can be >1m away from subject), compared to the HCW-swabbing process where a HCW is <1m away and face-to-face with the subject.

## Conclusion

This study demonstrates that while self-collection methods have a sensitivity of approximately 75%, it is inferior to the rate obtained by the health care worker administered swab (83.8%). The sensitivity of the self-collection methods is, however, higher and correlates better when Ct values of the HCW swabs are less than 30. The combined results of the saliva and self-swab test achieve a sensitivity equivalent to that of a health care worker administered swab. The specificity of the self-collection methods is 100%. Together with high specificity, we postulate that self-collection methods have their roles in diagnosis in early disease, where the viral load, and infectivity is high.

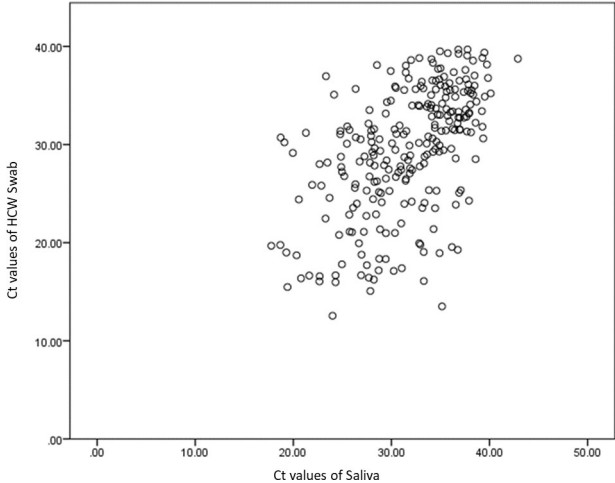

**Fig 2. Correlation of Ct values of HCW swab and saliva.**

**Table 7. Distribution of Ct values of the HCW swab, self-swab and saliva.**

| Test | Median (IQR*) of Ct value |
|---|---|
| HCW Swab | 31.59 (26.77, 35.62) |
| Self-swab | 31.65 (26.65, 35.94) |
| Saliva | 33.10 (28.25, 36.23) |

* IQR = Interquartile Range

## Supporting information

**S1 File. Provisional protocol for saliva sample collected in Lucence SAFER kit.**
(PDF)

**S2 File. Study protocol.**
(PDF)

**S3 File. Table with Ct values of N gene.**
(XLSX)

## Acknowledgments

We thank all clinical, nursing and allied health staff who provided care for the patients at Changi General Hospital, and Community Care Facility @ EXPO; staff in the Changi General Hospital Clinical Trials & Research Unit for coordinating patient recruitment, logistics management and assistance.

## Author Contributions

**Conceptualization:** Seow Yen Tan, Hong Liang Tey, Ernest Tian Hong Lim, Song Tar Toh, Yiong Huak Chan, Sing Ai Lee, Cheryl Xiaotong Tan, Gerald Choon Huat Koh, Thean Yen Tan, Chuin Siau.

**Data curation:** Seow Yen Tan, Hong Liang Tey, Ernest Tian Hong Lim, Yiong Huak Chan.

**Formal analysis:** Seow Yen Tan, Hong Liang Tey, Song Tar Toh, Yiong Huak Chan, Pei Ting Tan, Gerald Choon Huat Koh.

**Funding acquisition:** Sing Ai Lee, Cheryl Xiaotong Tan.

**Investigation:** Seow Yen Tan, Hong Liang Tey, Ernest Tian Hong Lim, Song Tar Toh, Gerald Choon Huat Koh, Thean Yen Tan, Chuin Siau.

**Methodology:** Seow Yen Tan, Hong Liang Tey, Song Tar Toh, Yiong Huak Chan, Gerald Choon Huat Koh, Thean Yen Tan, Chuin Siau.

**Project administration:** Seow Yen Tan, Hong Liang Tey, Ernest Tian Hong Lim, Song Tar Toh, Pei Ting Tan, Chuin Siau.

**Resources:** Ernest Tian Hong Lim, Song Tar Toh, Sing Ai Lee, Cheryl Xiaotong Tan, Chuin Siau.

**Supervision:** Hong Liang Tey, Ernest Tian Hong Lim, Song Tar Toh, Gerald Choon Huat Koh, Thean Yen Tan, Chuin Siau.

**Validation:** Seow Yen Tan, Gerald Choon Huat Koh.

**Visualization:** Sing Ai Lee, Cheryl Xiaotong Tan, Chuin Siau.

**Writing – original draft:** Seow Yen Tan, Hong Liang Tey, Ernest Tian Hong Lim, Song Tar Toh, Yiong Huak Chan.

**Writing – review & editing:** Seow Yen Tan, Hong Liang Tey, Ernest Tian Hong Lim, Song Tar Toh, Yiong Huak Chan, Pei Ting Tan, Sing Ai Lee, Cheryl Xiaotong Tan, Gerald Choon Huat Koh, Thean Yen Tan, Chuin Siau.

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
