## [Decision Letter · Decision Letter 0]

14 Oct 2020

PONE-D-20-28601

The Accuracy of Healthcare Worker versus Self Collected (2-in-1) Oropharyngeal and Bilateral Mid-Turbinate (OPMT) Swabs and Saliva Samples for SARS-CoV-2

PLOS ONE

Dear Dr. Tan,

Your manuscript has been reviewed by five experts in the field and their comments follow. All five reviewers like your work, but they also raised concerns and made specific suggestions. Addressing them in a revised paper will significantly improve the presentation and technical quality of your work. 

We look forward to receiving your revised manuscript.

Kind regards,

Dong-Yan Jin

Academic Editor

PLOS ONE

Journal Requirements:

2. In your Methods section, please provide additional information about the participant recruitment method and the demographic details of your participants. Please ensure you have provided sufficient details to replicate the analyses such as:

a) the recruitment date range (month and year),

b) a description of any inclusion/exclusion criteria that were applied to participant recruitment,

c) a table of relevant demographic details,

d) a statement as to whether your sample can be considered representative of a larger population, and

e) a description of how participants were recruited.

3. We note that you reference supplementary materials in your manuscript but there are no supplementary files attached. Please upload your supplementary files.

4. Thank you for providing the following Funding Statement: 

'This study was funded by Sheares Healthcare Group Pte Ltd. The funders had no role in study design, data collection and analysis, decision to publish, or preparation of the manuscript.'

a. We note that one or more of the authors is affiliated with the funding organization, indicating the funder may have had some role in the design, data collection, analysis or preparation of your manuscript for publication; in other words, the funder played an indirect role through the participation of the co-authors.

If the funding organization did not play a role in the study design, data collection and analysis, decision to publish, or preparation of the manuscript and only provided financial support in the form of authors' salaries and/or research materials, please review your statements relating to the author contributions, and ensure you have specifically and accurately indicated the role(s) that these authors had in your study in the Author Contributions section of the online submission form. Please make any necessary amendments directly within this section of the online submission form.

If the funding organization did have an additional role, please state and explain that role within your Funding Statement.

b. We also note that one or more of the authors are employed by another commercial company: Temasek Holdings

Please also declare this commercial affiliation, in your amended Funding Statement as well as a statement regarding the Role of Funders in your study. If the funding organization did not play a role in the study design, data collection and analysis, decision to publish, or preparation of the manuscript and only provided financial support in the form of authors' salaries and/or research materials, please review your statements relating to the author contributions, and ensure you have specifically and accurately indicated the role(s) that these authors had in your study. You can update author roles in the Author Contributions section of the online submission form.

c. Please also provide an updated Competing Interests Statement declaring these commercial affiliations along with any other relevant declarations relating to employment, consultancy, patents, products in development, or marketed products, etc.  

Within your Competing Interests Statement, please confirm that this commercial affiliation does not alter your adherence to all PLOS ONE policies on sharing data and materials by including the following statement: "This does not alter our adherence to  PLOS ONE policies on sharing data and materials.” (as detailed online in our guide for authors http://journals.plos.org/plosone/s/competing-interests). If this adherence statement is not accurate and  there are restrictions on sharing of data and/or materials, please state these.

Please note that we cannot proceed with consideration of your article until this information has been declared.

Additional Editor Comments:

Please make your best effort to address the reviewers' concerns.

Reviewers' comments:

Reviewer's Responses to Questions

**Comments to the Author**

1. Is the manuscript technically sound, and do the data support the conclusions?

Reviewer #1: Yes

Reviewer #2: Partly

Reviewer #3: Yes

Reviewer #4: Yes

Reviewer #5: Partly

2. Has the statistical analysis been performed appropriately and rigorously? 

Reviewer #1: Yes

Reviewer #2: I Don't Know

Reviewer #3: Yes

Reviewer #4: Yes

Reviewer #5: Yes

3. Have the authors made all data underlying the findings in their manuscript fully available?

Reviewer #1: No

Reviewer #2: Yes

Reviewer #3: Yes

Reviewer #4: No

Reviewer #5: Yes

4. Is the manuscript presented in an intelligible fashion and written in standard English?

Reviewer #1: Yes

Reviewer #2: No

Reviewer #3: Yes

Reviewer #4: Yes

Reviewer #5: Yes

5. Review Comments to the Author

Reviewer #1: In this study, Tan and colleagues present a comparison of self-collected swabs/ saliva vs. healthcare worker collected swabs. This is a controversial topic and worthy of exploration. Importantly, they find that self-collected options may be less sensitive than HCW collected swabs. The study is nicely performed and presented. It is particularly nice to see sample size analysis, which adds confidence in the conclusions. However, I do have a few recommendations to the authors to extract as much as possible from their data. I hope this helps them improve their manuscript further.

1. Please expand the abbreviation ‘OPMT’ when it first appears in the introduction.

2. Line 89: add a note on how patients in group 1 were confirmed to have COVID-19.

3. Can I confirm that the patient swabbed both mid-turbinates using a single swab stick and then put the same swab stick into their throat?

4. It is not clear what is meant by the ‘spitting’ method? Is this posterior oropharyngeal throat saliva collection? Collecting oral fluid, spitting out saliva, and collecting posterior oropharyngeal throat saliva are all likely to have different sensitivities for SARS-CoV-2 detection, so we need to define what exactly is being collected. Would actually be nice if the instructional videos could be uploaded as supplementary material.

5. Was there any particular timing of saliva collection? There is a tendency for early morning saliva to have higher viral loads (Hung DL et al, Open Forum Infect Dis, 2020).

6. Line 130 – 131: to clarify, the sample size was calculated based on a type I error rate of 1%?

7. Line 150: redundant ‘who’

8. Table 2 is not particularly useful and duplicates text in line 167 - 168. Could consider replacing with a contingency table of HCW swab vs saliva and HCW swab vs self-swab. This way, we can also check the % agreement and how many samples were detected by saliva/ self-swab, but not HCW swab.

9. Could add a McNemar test to compare sensitivities of saliva, self-swab and self-swab + saliva against the HCW-swab ‘gold standard’.

10. Table 3, 4: why is the total number of samples 336 (table 3), 335 (table 4)? The number of samples should be 401 – 27 (no. of negative samples) = 374? Are there missing data points?

11. Consider showing a scatterplot of the correlation of Ct values between self-swab and HCW swab and saliva and HCW swab.

12. Could include a column scatter plot comparing the RT-PCR Ct values of HCW swab, self-swab and saliva and statistically compare (? median) Cts of self-collected sample types to the HCW swab.

Reviewer #2: It is a great topic to compare the performance of self collected and HCW collected samples. But, the authors did not mention what clinical samples were used as gold standard for the diagnosis of COVID-19 in the 401 subjects. If HCW OP and MT swabs by HCW were used as gold standard, like procedure 2, listed under "test procedures", why the authors need to have a 2 stage design and include those previously tested positive for COVID-19 as subject and the second group of healthy volunteers as control? Why not include all the people when they were first tested for COVID-19 and test them with the three samples types at the very beginning?

The authors used the term 'detection rates' and 'negative correctness', are they referring to sensitivity and specificity which are more professional terms?

Reviewer #3: Tan SY and colleagues performed a cross-sectional study to investigate healthcare worker vs. self-collected OPMT swabs and saliva samples for the detection of SARS-CoV-2 among persons with a confirmed diagnosis of COVID-19 and healthy volunteers.

I have some concerns as follow:

Major comments:

1. Characteristics, e.g. age, the onset of symptoms, severity of the disease, of the study populations should be included to provide readers to understand the clinical setting of the study better. The different settings may associate with different sensitivity of each specimen. One study demonstrated the lower value of saliva for the diagnosis of COVID-19 in children (Chong CY, et al. Clin Infect Dis 2020; article in press). Many studies showed more testing agreement of saliva and nasopharyngeal swab at the earlier onset of the disease (Jamal AJ, et al. Clin Infect Dis 2020; article in press, Iwasaki S, et al. J Infect 2020; article in press).

2. The gene that was RT-PCR test should be described. Was a housekeeping gene included in the RT-PCR reaction? The presence of the housekeeping gene in the RT-PCR test could help to determine the adequacy of a specimen collection.

3. In the first paragraph of the discussion, the author stated: “Our findings corroborate with existing epidemiologic data which indicates that while viral RNA detection may persist in some patients, such persistent RNA detection likely represents non-viable virus and hence, such patients are noninfectious.” The author should demonstrate the results in their study that suggest the conclusion of this statement.

Minor comments:

1. The laboratory processing method that resulted in a lower yield of the saliva should be discussed. In this work, saliva samples were collected using the SAFER-Sample, which a saliva solubilizing solution was added to the samples. A preprint by Griesemer SB and coworkers demonstrated lower sensitivity to detect the virus when saliva stabilizing solution was added. Some studies showed a higher sensitivity of saliva when compare to nasopharyngeal swab. In these studies, the authors did not put either saliva stabilizing solution or viral transport media in the saliva specimen (Wylle AL, et al. N Engl J Med 2020; article in press; Rao M, et al. Clin Infect Dis 2020; article in press).

2. Please update the references. The preprints were accepted for publication.

3. The writing can be improved by reorganizing the content to increase the continuity of the idea and content in the manuscript.

Reviewer #4: The manuscript by Tan et al , compared the accuracy of healthcare worker versus self collected OPMT swabs and saliva samples for SARS-CoV-2. This is a comprehensive and important study during this COVID-19 pandemic.

I have the following minor comments:

- the authors may consider adding a flowchart for the workflow and sample size of the study

- the authors should include supplementary file for all single Ct values of different samples matching each patients

Reviewer #5: In the manuscript entitled “The Accuracy of Healthcare Worker versus Self Collected (2-in-1) Oropharyngeal and Bilateral Mid-Turbinate (OPMT) Swabs and Saliva Samples for SARS-CoV-2”, Seow Yen Tan et al compares the detection rate of SARS-C0V-2 by RT-PCR in self- collected samples and samples collected by HCW. In addition, they used saliva, OP swabs and both methods combined, and compared the sensitivity of each approach in 400 samples from patients diagnosed with COVID-19 and 100 negative subjects. They concluded that saliva and sel-collected samples are inferior to OP and HCW-collected ones, respectively. However, combining both self-collected samples provided a higher detection rate.

Some points to discuss:

1. How long it took to test saliva samples after collecting them? Although swab samples were preserved in transportation medium, saliva could be affected by time until testing;

2. Collecting saliva samples as the last procedure could affect results? Self-collected swab, followed by HCW collected swabs could interfere in the quantity and quality of saliva samples;

3. Line 167 “self-saliva” means “self-swab”?? Please, correct it;

4. In several parts of text authors refer to testing in “early phase” would provide better results: how early??In addition, there is no description on details of methodology, like time to test, preservation of samples, and characteristics of patients with positive and negative results. What about the time since diagnosis or duration of symptoms? Or severity of disease? These are important information to understand how samples were selected, and how such information could help explaining the results obtained;

5. Authors state at conclusion that self-collection would be preferable for use in low prevalence population? Why??

6. How author explain their results are so distinct of other reports, regarding sensitivity of saliva testing? It should be addressed in discussion. In limitations of study they cite the use of a very specific population (male migrants). How could this fact impact the results?

Minor comments: there are several typos and gramatical mistakes. A deeper English revision is warranted.

6. PLOS authors have the option to publish the peer review history of their article (what does this mean?). If published, this will include your full peer review and any attached files.

Reviewer #1: **Yes: **Siddharth Sridhar

Reviewer #2: No

Reviewer #3: No

Reviewer #4: No

Reviewer #5: No

---

## [Author Response · Author response to Decision Letter 0]

26 Nov 2020

Dear Editor,

Thank you for consideration of our manuscript. We have made considerable changes and will address your comments as well as the reviewers’ comments (in bold), point by point below.

Have looked through the pages and the files seem to be fulfilling the requirement. Hope they are acceptable. 

2. In your Methods section, please provide additional information about the participant recruitment method and the demographic details of your participants. Please ensure you have provided sufficient details to replicate the analyses such as:

a) the recruitment date range (month and year),

b) a description of any inclusion/exclusion criteria that were applied to participant recruitment,

c) a table of relevant demographic details,

d) a statement as to whether your sample can be considered representative of a larger population, and

e) a description of how participants were recruited.

The requested information have been included in the revised manuscript

3. We note that you reference supplementary materials in your manuscript but there are no supplementary files attached. Please upload your supplementary files.

Supplementary files have been uploaded. Apologies for missing them out in the initial submission.

4. Comment regarding affiliation of authors SAL and CXT to commercial entities.

I am unable to find the column to enter the amended funding statement in the entire revision process, hence the amended funding statement has been indicated in the cover letter as instructed by the video guide. The amended funding statement reads:

"This study was funded by Sheares Healthcare Group Pte Ltd. 

The funder provided support in the form of salaries for authors SAL and CXT, but did not have any additional role in the study design, data collection and analysis, decision to publish, or preparation of the manuscript. The specific roles of these authors are articulated in the ‘author contributions’ section.

This does not alter our adherence to PLOS ONE policies on sharing data and materials.

Besides that, author CXT is employed by Temasek Holdings, and was acting on behalf of Sheares Healthcare Group Pte Ltd for the study. Temasek Holdings did not have any additional role in the study design, data collection and analysis, decision to publish, or preparation of the manuscript. This commercial affiliation does not alter our adherence to PLOS ONE policies on sharing data and materials."

Reviewer #1: In this study, Tan and colleagues present a comparison of self-collected swabs/ saliva vs. healthcare worker collected swabs. This is a controversial topic and worthy of exploration. Importantly, they find that self-collected options may be less sensitive than HCW collected swabs. The study is nicely performed and presented. It is particularly nice to see sample size analysis, which adds confidence in the conclusions. However, I do have a few recommendations to the authors to extract as much as possible from their data. I hope this helps them improve their manuscript further.

1. Please expand the abbreviation ‘OPMT’ when it first appears in the introduction.

- Abbreviation expanded as per recommendation

 2. Line 89: add a note on how patients in group 1 were confirmed to have COVID-19.

- The sentence ‘Diagnosis of COVID-19 was confirmed via a positive RT-PCR from a nasopharyngeal swab’ was added.

 3. Can I confirm that the patient swabbed both mid-turbinates using a single swab stick and then put the same swab stick into their throat?

- Correct, the same stick is used, but first by swabbing the throat, then the bilateral mid-turbinates. 

 4. It is not clear what is meant by the ‘spitting’ method? Is this posterior oropharyngeal throat saliva collection? Collecting oral fluid, spitting out saliva, and collecting posterior oropharyngeal throat saliva are all likely to have different sensitivities for SARS-CoV-2 detection, so we need to define what exactly is being collected. Would actually be nice if the instructional videos could be uploaded as supplementary material.

- We collected posterior oropharyngeal throat saliva, which we are now calling it deep throat saliva, which we hope would clarify the nature of the saliva that is collected. We are currently unable to upload the instructional video as the producer has not given the permission to upload it as a supplementary material.

 5. Was there any particular timing of saliva collection? There is a tendency for early morning saliva to have higher viral loads (Hung DL et al, Open Forum Infect Dis, 2020).

- There was no particular timing for saliva collection. The only restriction we placed was that the subject should not have any food/drink 30 minutes prior to the saliva collection. We decided not to limit to early morning saliva, as when the test is being used on a large scale, it is likely that patients will be tested at any time of the day. Hence the study design to simulate a real life situation. 

 6. Line 130 – 131: to clarify, the sample size was calculated based on a type I error rate of 1%?

For clarity, we have changed ' An error rate of less than 1% was determined to be of clinical relevance so a sample size of at least 400 subjects was calculated' to ‘postulating a 100% accuracy , 400 subjects will be required to achieve a lower 95% Confidence interval 99.1% (which gives a less than 1% error rate)’

7. Line 150: redundant ‘who’

The word “who” deleted

8. Table 2 is not particularly useful and duplicates text in line 167 - 168. Could consider replacing with a contingency table of HCW swab vs saliva and HCW swab vs self-swab. This way, we can also check the % agreement and how many samples were detected by saliva/ self-swab, but not HCW swab.

Added table to address the comments in both Q8 and Q9

 9. Could add a McNemar test to compare sensitivities of saliva, self-swab and self-swab + saliva against the HCW-swab ‘gold standard’.

Added table to address the comments in both Q8 and Q9

10. Table 3, 4: why is the total number of samples 336 (table 3), 335 (table 4)? The number of samples should be 401 – 27 (no. of negative samples) = 374? Are there missing data points?

The original Table 3 and 4 essentially captures all those that were positive of both tests. Table 3 contains the 336 subjects where both the self-swab and HCW swab were positive, and Table 4 contains the 335 that were positive on both the saliva and HCW swab.

 11. Consider showing a scatterplot of the correlation of Ct values between self-swab and HCW swab and saliva and HCW swab.

Have added a scatterplot to show the correlation of the Ct values of the HCW swab and self-swab, and HCW swab and saliva. 

 12. Could include a column scatter plot comparing the RT-PCR Ct values of HCW swab, self-swab and saliva and statistically compare (? median) Cts of self-collected sample types to the HCW swab.

A table showing the distribution of the Ct values in the 3 tests were included as Table 7. Using Wilcoxon Signed Rank Test, the difference in CT values between self-swab and HCW swab is statistically significance, p=0.026 similarly for the saliva and HCW swab, p<0.001. The self-swab correlates better with the HCW swab. 

Reviewer #2: 

It is a great topic to compare the performance of self collected and HCW collected samples. But, the authors did not mention what clinical samples were used as gold standard for the diagnosis of COVID-19 in the 401 subjects. If HCW OP and MT swabs by HCW were used as gold standard, like procedure 2, listed under "test procedures", why the authors need to have a 2 stage design and include those previously tested positive for COVID-19 as subject and the second group of healthy volunteers as control? Why not include all the people when they were first tested for COVID-19 and test them with the three samples types at the very beginning?

The diagnosis of COVID-19 in this population was made based on a positive RT-PCR on a nasopharyngeal swab. During the study period in Singapore, all patients with COVID-19 are kept isolated for at least 21 days, before returning to the community. At that point in time, where the prevalence was not high in the nation, to achieve the desired sample size of 400, we would need to sample a much larger number. Hence we elected to recruit the patients that are admitted to the community care facility and in one of the public hospital. Recognizing that with this population, we would not be able to assess the specificity of the self-swab and saliva testing, hence the second group of healthy volunteers. This approach has allowed us to be assess both the sensitivity and specificity adequately, without too high a sampling burden. 

The authors used the term 'detection rates' and 'negative correctness', are they referring to sensitivity and specificity which are more professional terms?

Duly noted on the comment. Have changed ‘detection rates’ and ‘negative correctness’ to ‘sensitivity’ and ‘specificity’ respectively. 

Reviewer #3: Tan SY and colleagues performed a cross-sectional study to investigate healthcare worker vs. self-collected OPMT swabs and saliva samples for the detection of SARS-CoV-2 among persons with a confirmed diagnosis of COVID-19 and healthy volunteers.

 I have some concerns as follow:

 Major comments:

1. Characteristics, e.g. age, the onset of symptoms, severity of the disease, of the study populations should be included to provide readers to understand the clinical setting of the study better. The different settings may associate with different sensitivity of each specimen. One study demonstrated the lower value of saliva for the diagnosis of COVID-19 in children (Chong CY, et al. Clin Infect Dis 2020; article in press). Many studies showed more testing agreement of saliva and nasopharyngeal swab at the earlier onset of the disease (Jamal AJ, et al. Clin Infect Dis 2020; article in press, Iwasaki S, et al. J Infect 2020; article in press).

We have added a table on the demographics of the study subjects. However, the data that we have is not complete, as some data such as date of birth could not be given to the study due to regulations from the study site. The subjects generally had mild disease, with those who are symptomatic had only upper respiratory tract symptoms, without any need for supplemental oxygen. Most subjects were within 7 days of illness onset or first positive swab. 

2. The gene that was RT-PCR test should be described. Was a housekeeping gene included in the RT-PCR reaction? The presence of the housekeeping gene in the RT-PCR test could help to determine the adequacy of a specimen collection.

The gene that was tested was the ‘N’ gene and the ‘ORF1ab’ gene. The Ct value from the ‘N’ gene was used in the statistical analysis involving Ct values. There is an internal control that is being run is each test. 

 3. In the first paragraph of the discussion, the author stated: “Our findings corroborate with existing epidemiologic data which indicates that while viral RNA detection may persist in some patients, such persistent RNA detection likely represents non-viable virus and hence, such patients are noninfectious.” The author should demonstrate the results in their study that suggest the conclusion of this statement.

Viral cultures were not done for this study. This statement was made based on the position paper from the Academy of Medicine, Singapore as well as data from a local study (References number 9 and 10) that above the Ct values of 30, SARS CoV 2 could not be cultured. Nonetheless, as out study did not involve viral cultures, we have removed this statement from the manuscript. 

Minor comments:

1. The laboratory processing method that resulted in a lower yield of the saliva should be discussed. In this work, saliva samples were collected using the SAFER-Sample, which a saliva solubilizing solution was added to the samples. A preprint by Griesemer SB and coworkers demonstrated lower sensitivity to detect the virus when saliva stabilizing solution was added. Some studies showed a higher sensitivity of saliva when compare to nasopharyngeal swab. In these studies, the authors did not put either saliva stabilizing solution or viral transport media in the saliva specimen (Wylle AL, et al. N Engl J Med 2020; article in press; Rao M, et al. Clin Infect Dis 2020; article in press).

The evidence that have emerged since our study was conducted have indicated that neat samples have given a better yield, as indicated by your good self. The initial concern was that of biosafety, and SAFER Sample was one of the first saliva collection kits to be evaluated locally when the study was being planned. We have added the discussion point that the stabilising solution might have had a negative impact on the yield of saliva testing. 

2. Please update the references. The preprints were accepted for publication.

References number 3 and 8 updated. 

3. The writing can be improved by reorganizing the content to increase the continuity of the idea and content in the manuscript.

e

Revised as suggested

Reviewer #4: The manuscript by Tan et al , compared the accuracy of healthcare worker versus self collected OPMT swabs and saliva samples for SARS-CoV-2. This is a comprehensive and important study during this COVID-19 pandemic.

 I have the following minor comments:

 - the authors may consider adding a flowchart for the workflow and sample size of the study

The workflow of the study process was added to the manuscript. The sample size was not included in the flowchart as there was no dropout from the study per se, but there was 1 subject that could not provide a saliva sample, hence the additional recruitment of 1 subject, giving a total number of 401 COVID-19 positive subjects.

 - the authors should include supplementary file for all single Ct values of different samples matching each patients

Added this into the supplementary material as suggested.

Reviewer #5:

 In the manuscript entitled “The Accuracy of Healthcare Worker versus Self Collected (2-in-1) Oropharyngeal and Bilateral Mid-Turbinate (OPMT) Swabs and Saliva Samples for SARS-CoV-2”, Seow Yen Tan et al compares the detection rate of SARS-C0V-2 by RT-PCR in self- collected samples and samples collected by HCW. In addition, they used saliva, OP swabs and both methods combined, and compared the sensitivity of each approach in 400 samples from patients diagnosed with COVID-19 and 100 negative subjects. They concluded that saliva and sel-collected samples are inferior to OP and HCW-collected ones, respectively. However, combining both self-collected samples provided a higher detection rate.

 Some points to discuss:

1. How long it took to test saliva samples after collecting them? Although swab samples were preserved in transportation medium, saliva could be affected by time until testing;

All samples (swabs and saliva) were processed within 24 hours of collection. The samples were stored at 2-8°C. Based on the recommendations provided by the manufacturers of SAFER-sample, the samples are stable for 72 hours.

2. Collecting saliva samples as the last procedure could affect results? Self-collected swab, followed by HCW collected swabs could interfere in the quantity and quality of saliva samples;

We determined the sequence of testing as such, thinking that doing the swabs first, will affect the saliva yield to a lesser degree, when compared to doing saliva collection first then the swabs. This is due to the theoretical possibility that by expectorating deep throat saliva, it would potentially reduce the presence of viral material.

 3. Line 167 “self-saliva” means “self-swab”?? Please, correct it;

Typographical error corrected

 4. In several parts of text authors refer to testing in “early phase” would provide better results: how early??In addition, there is no description on details of methodology, like time to test, preservation of samples, and characteristics of patients with positive and negative results. What about the time since diagnosis or duration of symptoms? Or severity of disease? These are important information to understand how samples were selected, and how such information could help explaining the results obtained; 

The above information were added in the revised manuscript. 

5. Authors state at conclusion that self-collection would be preferable for use in low prevalence population? Why?? 

We have modified the phrasing of the statement. The thought was that in the situations where asymptomatic infection is occurring, and mass surveillance is being undertaken, an enormous pool of trained swabbers is needed to carry this out. Hence self-collection methods would be useful in situations when number of trained swabbers need to be scaled up rapidly, without a corresponding jump in infectious exposure to the staff. 

6. How author explain their results are so distinct of other reports, regarding sensitivity of saliva testing? It should be addressed in discussion. In limitations of study they cite the use of a very specific population (male migrants). How could this fact impact the results?

This could be due to the fact that the sampling process is unique that the study design was that the subjects were already known to be positive for COVID-19. Hence the way the analysis is conducted appears unique. Most other studies would be recruiting from a testing centres where the status of the subjects are unknown and the swab done by a healthcare worker is deemed to be the gold standard. Have added this into the discussion section. 

The study population is certainly not generalizable to the local population. Nonetheless, the self-collection methods is meant to be as simple as possible so that the vast majority in the population is able to follow the instructions and provide a good sample. 

 Minor comments: there are several typos and gramatical mistakes. A deeper English revision is warranted.

Revision done as advised.

---

## [Decision Letter · Decision Letter 1]

7 Dec 2020

PONE-D-20-28601R1

The Accuracy of Healthcare Worker versus Self Collected (2-in-1) Oropharyngeal and Bilateral Mid-Turbinate (OPMT) Swabs and Saliva Samples for SARS-CoV-2

PLOS ONE

Dear Dr. Tan,

Thank you for submitting your revised manuscript to PLOS ONE.

We have now received comments from the original reviewers. Two of them carefully picked up some typos and errors in your paper that must be corrected in the final version. 

After careful consideration, we feel that your paper has merit but does not fully meet PLOS ONE’s publication criteria as it currently stands. Therefore, we invite you to submit a revised version of the manuscript that addresses the points raised during the review process.

We look forward to receiving your revised manuscript.

Kind regards,

Dong-Yan Jin

Academic Editor

PLOS ONE

Reviewers' comments:

Reviewer's Responses to Questions

**Comments to the Author**

1. If the authors have adequately addressed your comments raised in a previous round of review and you feel that this manuscript is now acceptable for publication, you may indicate that here to bypass the “Comments to the Author” section, enter your conflict of interest statement in the “Confidential to Editor” section, and submit your "Accept" recommendation.

Reviewer #1: All comments have been addressed

Reviewer #3: All comments have been addressed

Reviewer #4: All comments have been addressed

Reviewer #5: All comments have been addressed

2. Is the manuscript technically sound, and do the data support the conclusions?

Reviewer #1: Yes

Reviewer #3: Partly

Reviewer #4: Yes

Reviewer #5: Yes

3. Has the statistical analysis been performed appropriately and rigorously? 

Reviewer #1: Yes

Reviewer #3: Yes

Reviewer #4: Yes

Reviewer #5: Yes

4. Have the authors made all data underlying the findings in their manuscript fully available?

Reviewer #1: Yes

Reviewer #3: Yes

Reviewer #4: Yes

Reviewer #5: Yes

5. Is the manuscript presented in an intelligible fashion and written in standard English?

Reviewer #1: Yes

Reviewer #3: (No Response)

Reviewer #4: Yes

Reviewer #5: Yes

6. Review Comments to the Author

Reviewer #1: Just a few minor points:

1. Correct grammar for this sentence: “Another study on 236 ambulatory, literate, mostly adult subjects the performance of self-collected nasal and throat swabs was at least equivalent to that of health worker collected swabs…”

2. Correct grammar for this sentence: “Self-collection of samples would reduce very significantly on the reliance of trained personnel to collect samples and ramp up testing capacity.”

3. Clarify in the methods that subjects were supervised during self-sample collection. This is hinted at in the discussion.

4. Line 155: inappropriately italicized ‘R’ in reverse-transcription.

5. Line 163: SARS-CoV-2, not SARS-Cov-2.

6. Line 265: Please elaborate on this statement. I can understand why the study findings would not be applicable to pediatric and very elderly populations, but the study findings should be broadly applicable to other sections of Singapore’s population?

Reviewer #3: My comments have been addressed.

However, some typos are still noted, e.g., Lines 246 and 247: the legends of figures 1 and 2 describe the same thing. The manuscript might be benifit from language editing.

In the first paragraph of the discussion, the authors stated the study on patients who are already known to be COVID-19 as the strength of the study. I think this is more likely to be the limitation, as the mean duration of the first positive swab to the study day and the mean duration between illness onset to study day were quite long. It is well known that Ct value correlated with days from symptom onset. Sensitivity of the virus detection is decreasing in saliva collected from later time of illness onset.

Ref: Jamal AJ, et al. Clin Infect Dis. 2020;ciaa848. Williams E, et al. J Clin Micriobiol. 2020;58(8):e00776-20.

Reviewer #4: The authors have addressed all my concerns, i have no further questions. The manuscript should be ready to published.

Reviewer #5: (No Response)

7. PLOS authors have the option to publish the peer review history of their article (what does this mean?). If published, this will include your full peer review and any attached files.

Reviewer #1: **Yes: **Siddharth Sridhar

Reviewer #3: No

Reviewer #4: No

Reviewer #5: **Yes: **Carlos Brites

---

## [Author Response · Author response to Decision Letter 1]

9 Dec 2020

Dear Reviewers, thank you for the kind comments and feedback. Kindly see the responses below to your comments. 

Reviewer #1: Just a few minor points:

1. Correct grammar for this sentence: “Another study on 236 ambulatory, literate, mostly adult subjects the performance of self-collected nasal and throat swabs was at least equivalent to that of health worker collected swabs…”

This sentence is corrected to: “In another study on 236 ambulatory subjects, the performance of self-collected nasal and throat swabs is at least equivalent to that of health worker collected swabs for the detection of SARS-CoV-2 and other respiratory viruses.”

2. Correct grammar for this sentence: “Self-collection of samples would reduce very significantly on the reliance of trained personnel to collect samples and ramp up testing capacity.”

This sentence is corrected to “If the self-collection of samples is proven to be a reliable alternative to a HCW swab, it would reduce the reliance of trained personnel to collect samples and enable a rapid increase in testing capacity. It would also reduce greatly the biosafety risk that is posed to HCWs and help with PPE conservation efforts.”

3. Clarify in the methods that subjects were supervised during self-sample collection. This is hinted at in the discussion.

Added in Line 145: “Study team members were present on site to observe and supervise the self-collection process”

4. Line 155: inappropriately italicized ‘R’ in reverse-transcription.

Thanks for pointing this out, this has been changed.

5. Line 163: SARS-CoV-2, not SARS-Cov-2.

Thanks for pointing this out, this has been changed.

6. Line 265: Please elaborate on this statement. I can understand why the study findings would not be applicable to pediatric and very elderly populations, but the study findings should be broadly applicable to other sections of Singapore’s population?

We made this statement as the demographic profile of the COVID-19 group (all males, generally young and healthy) is not representative of the demographics of the Singapore. 

I have edited it to read: “Hence the results from this study might not be applicable to the general population, as the paediatric and very elderly were not included in the study. The migrant worker population in this study, which consist of generally young and healthy males, is also not representative of the demographics of Singapore.”

Reviewer #3: My comments have been addressed.

However, some typos are still noted, e.g., Lines 246 and 247: the legends of figures 1 and 2 describe the same thing. The manuscript might be benifit from language editing.

My apologies for the error, thank you for pointing this out. The legend is corrected, to reflect:

Fig 1: Correlation of Ct values of HCW swab and self-swab

Fig 2: Correlation of Ct values of HCW swab and saliva

The other typographical errors have been corrected to the best of our ability.

In the first paragraph of the discussion, the authors stated the study on patients who are already known to be COVID-19 as the strength of the study. I think this is more likely to be the limitation, as the mean duration of the first positive swab to the study day and the mean duration between illness onset to study day were quite long. It is well known that Ct value correlated with days from symptom onset. Sensitivity of the virus detection is decreasing in saliva collected from later time of illness onset.

Ref: Jamal AJ, et al. Clin Infect Dis. 2020;ciaa848. Williams E, et al. J Clin Micriobiol. 2020;58(8):e00776-20.

Agree with your point that the mean duration of the first positive swab to the study day and the mean duration between illness onset to study day were quite long hence the study design itself might have led to the results that we are seeing, where the yield of the saliva is not as good as the HCW swab. Our thought was that there was a large number of subjects who were known to be COVID-19 positive in our study, hence the perception of this being a strength. 

I have restructured the discussion to improve the flow of the manuscript. The section pertaining to your comments above would read:

“The strength of our study is the large number of subjects confirmed to have COVID-19. Besides that, the study also included a high proportion of asymptomatic individuals who were picked up because of Singapore’s proactive mass screening policy. The combination of self-swab and saliva sampling performed well in these asymptomatic subjects, implying that the strategy of combined self-testing, has the ability diagnose COVID-19 in asymptomatic individuals with a sensitivity equivalent to that of a swab by a HCW. The study results from the healthy volunteers indicate a low false positive rate with self-collection methods. 

These findings, indicate that self-collection methods may be a useful tool for COVID-19 surveillance in the asymptomatic individuals, and in situations where testing capacity needs to be scaled up rapidly, without a need for large increase of manpower, and without increased infectious exposure to the swabbing staff. Testing strategies can be tailored based on the target population and the intended use of the various tests on its own or in combination. 

The way the study findings were presented are unlike most studies involving saliva testing for COVID-19. This is probably due to the fact that our study is carried out on subjects who are already known to have COVID-19, unlike most studies which are done in testing centres where the potential subjects’ results are still unknown. This also meant that the sampling was done later in the subjects’ trajectory of illness, as they were first tested positive for COVID-19, then enrolled into the study. The later sampling possibly had a negative impact on the sensitivity of the saliva [11].”

Reviewer #4: The authors have addressed all my concerns, i have no further questions. The manuscript should be ready to published.

Reviewer #5: (No Response)

---

## [Editor Report · Decision Letter 2]

10 Dec 2020

The Accuracy of Healthcare Worker versus Self Collected (2-in-1) Oropharyngeal and Bilateral Mid-Turbinate (OPMT) Swabs and Saliva Samples for SARS-CoV-2

PONE-D-20-28601R2

Dear Dr. Tan,

We’re pleased to inform you that your manuscript has been judged scientifically suitable for publication and will be formally accepted for publication once it meets all outstanding technical requirements.

Kind regards,

Dong-Yan Jin

Academic Editor

PLOS ONE
---

## [Editor Report · Acceptance letter]

14 Dec 2020

PONE-D-20-28601R2 

The Accuracy of Healthcare Worker versus Self Collected (2-in-1) Oropharyngeal and Bilateral Mid-Turbinate (OPMT) Swabs and Saliva Samples for SARS-CoV-2 

Dear Dr. Tan:

I'm pleased to inform you that your manuscript has been deemed suitable for publication in PLOS ONE. Congratulations! Your manuscript is now with our production department. 

Kind regards, 

on behalf of

Professor Dong-Yan Jin 

Academic Editor

PLOS ONE